# A Study of R-R Interval Transition Matrix Features for Machine Learning Algorithms in AFib Detection

**DOI:** 10.3390/s23073700

**Published:** 2023-04-03

**Authors:** Sahil Patel, Maximilian Wang, Justin Guo, Georgia Smith, Cuixian Chen

**Affiliations:** 1John T. Hoggard High School, Wilmington, NC 28403, USA; 2Department of Mathematics and Statistics, University of North Carolina Wilmington, Wilmington, NC 28403, USA; 3Isaac M. Bear Early College High School, Wilmington, NC 28403, USA; 4Department of Computer Science, University of North Carolina at Chapel Hill, Chapel Hill, NC 27599, USA; 5Department of Biostatistics, University of Colorado’s Anschutz Medical Campus, Aurora, CO 80045, USA

**Keywords:** atrial fibrillation (AFib), R-R intervals, transition matrix, machine learning

## Abstract

Atrial Fibrillation (AFib) is a heart condition that occurs when electrophysiological malformations within heart tissues cause the atria to lose coordination with the ventricles, resulting in “irregularly irregular” heartbeats. Because symptoms are subtle and unpredictable, AFib diagnosis is often difficult or delayed. One possible solution is to build a system which predicts AFib based on the variability of R-R intervals (the distances between two R-peaks). This research aims to incorporate the transition matrix as a novel measure of R-R variability, while combining three segmentation schemes and two feature importance measures to systematically analyze the significance of individual features. The MIT-BIH dataset was first divided into three segmentation schemes, consisting of 5-s, 10-s, and 25-s subsets. In total, 21 various features, including the transition matrix features, were extracted from these subsets and used for the training of 11 machine learning classifiers. Next, permutation importance and tree-based feature importance calculations determined the most predictive features for each model. In summary, with Leave-One-Person-Out Cross Validation, classifiers under the 25-s segmentation scheme produced the best accuracies; specifically, Gradient Boosting (96.08%), Light Gradient Boosting (96.11%), and Extreme Gradient Boosting (96.30%). Among eleven classifiers, the three gradient boosting models and Random Forest exhibited the highest overall performance across all segmentation schemes. Moreover, the permutation and tree-based importance results demonstrated that the transition matrix features were most significant with longer subset lengths.

## 1. Introduction

### 1.1. Background

Atrial fibrillation (AFib) is a heart affliction characterized by “irregularly irregular” heartbeats [1]. During AFib, electrophysiological and/or structural malformations within heart tissues cause the atria to lose coordination with the ventricles [2,3]. AFib currently affects around 2.5 million Americans and 8.8 million Europeans, and while it is rare in children, it is extremely common in the elderly population, with a 1–3% presence in individuals greater than 20 years of age and a near-20% presence in individuals greater than 85 years of age [2,4,5]. AFib is also attributed to a higher morbidity and mortality rate due to concomitant diseases, such as neoplasm, chronic renal failure, and chronic obstructive pulmonary disease [6,7,8]. Other health consequences can include an increased risk of stroke, cognitive/memory impairment, and Alzheimer’s disease [7,8]. AFib is also linked to a lower quality of life, with patients’ quality of life being impaired at around 8 h of daily burden [9]. Furthermore, diagnosis and treatment of AFib is often delayed until it has progressed significantly in severity, at which point symptoms become noticeable [10]. As AFib is a progressive disease, any delay in diagnosis allows for further cumulative damage to the atria [11]. Because of the association between AFib and higher mortality and a lower quality of life, as well as the prolonged gap between patient affliction and diagnosis/treatment, it is critical to quickly, accurately, and reliably detect AFib in order to best mitigate health consequences [12].

Due to its random and spontaneous nature, AFib is difficult to identify and diagnose through routine medical examinations. Suspected AFib can be investigated with manual pulse rate and EKG analysis, but this is ineffective in diagnosing paroxysmal AFib, which typically requires a Holter monitor (24-h) or event monitor (7–30 day) recording, a physical examination, a complete blood count, transthoracic echocardiography, and/or chest radiography, varying from patient to patient [3]. Thus, a machine learning-based application that constantly tracks a user’s heartbeats and analyzes them for AFib would greatly facilitate the process of atrial fibrillation detection. This application could be implemented in wearable devices such as Apple Watches, allowing for convenient and accessible heartbeat monitoring [13,14].

R-R intervals, defined as the distance between the R-peaks of two consecutive heartbeats (shown in Figure 1), are frequently implemented in various methods of AFib detection [15,16,17]. This is because R-R interval variability is a strong indicator and effective diagnostic method for AFib [18]. The current medical standard involves identifying AFib through the lack of P-waves from EKGs [16,19,20,21]. Chaotic atrial activity has also been utilized to identify AFib [19,22,23,24]. Nevertheless, R-R interval variability features have been shown to be an effective discriminator of atrial fibrillation with low computational intensity [22,23].

### 1.2. Related Work

Research in atrial fibrillation detection using machine learning techniques has been performed for many decades. A 1999 study utilized a Hidden Markov Model (HMM) algorithm, along with other measures, for AFib detection, finding that the HMM outperformed the other features in terms of sensitivity and positive predictivity [25]. In 2001, Tateno and Glass created a method for detecting atrial fibrillation using the coefficients of variation and a Kolomogorov–Smirnov Test on density histograms of RR and ΔRR intervals [15]. Their best results were obtained using the Kolomogorov–Smirnov Test on ΔRR intervals, with a sensitivity of 94.4% and specificity of 97.2%. Additionally, Ghodrati and Marinello used two different probability density functions to model the histogram of R-R intervals and determined which density function led to more accurate AFib detection [26]. The best function resulted in a sensitivity of 89% and a positive predicative value (PPV) of 87% on the MIT-BIH AFib database.

More recent research in AFib detection has increasingly incorporated advanced machine learning techniques [27]. A 2020 study performed wavelet transformations on Lorenz plots and fed these transformed plots into support vector machines (SVM), yielding a sensitivity of 99.2% and a specificity of 99.5% [14]. Kennedy et al. utilized four different R-R irregularity measurements to train the Random Forest (RF) and K-Nearest Neighbors (KNN) classifiers [28] to predict AFib. The best sensitivity (97.6%) was obtained by the coefficient of Sample Entropy feature alone, while the highest median accuracy (99.1%), specificity (98.3%), and PPV (92.8%) were found when the RF Classifier was combined with the coefficient of Sample Entropy. Another study conducted by Hirsch et al. incorporated three machine learning models: Boosted Trees, Random Forest, and Linear Discriminant Analysis [23]. The RF Classifier outperformed the other models, with an accuracy, sensitivity, and specificity of 97.6%, 98.0%, and 97.4%, respectively. Furthermore, a step-by-step machine learning pipeline was created to feed over 300 features into the XGBoost Classifier [29], resulting in a cross-validation F1-Score of 82.45% on the 2017 PhysioNet Challenge dataset.

With the surge of interest in deep learning, researchers have explored the application of deep learning techniques for AFib detection [16,27,30,31,32,33]. Liaqat et al. employed machine learning classifiers, including SVM, Logistic Regression, and XGBoost, as well as deep learning models, such as convolutional neural networks (CNN) and long short-term memory (LSTM) for AFib detection [30]. Out of all models, LSTM obtained the highest accuracy on the MIT-BIH dataset, at 82.9%, while CNN yielded the best precision (82.0%), sensitivity (81.0%), and F1-score (82.6%). Krol-Jozaga used three methods to create 2D representations of EKG signals as inputs to CNNs: scalograms, spectrograms, and attractor reconstruction [33]. Sensitivities of 94%, 95%, and 90% were obtained using scalograms, spectrograms, and attractor reconstruction, respectively. Chen et al. utilized a feedforward neural network that achieved an accuracy, sensitivity, and specificity of 84.00%, 84.26%, and 93.23%, respectively, on a mixed dataset composed of the 2017 Challenge Dataset and MIT-BIH AFib Dataset [34]. Additionally, Nankani et al. investigated a transformer neural network, obtaining an F1-Score of 0.87 on the 2017 Challenge Dataset [35].

Concerns have been raised about the intransparency of deep neural networks, which makes it challenging for medical professionals to obtain useful information regarding the AFib classification. [27]. Many methods to improve the transparency of neural networks have been proposed, allowing the classification results to be more informative for individuals and medical professionals [36,37]

Rather than evaluating the most effective overall classifiers, including deep learning models, the primary focus of our study is to identify the most significant features when combined with lightweight and accessible machine learning (ML) classifiers. The purpose of using lightweight ML classifiers over deep learning methods is to lower computational intensity while still maintaining relatively high classification performance. This is especially important for applications in wearable devices.

Furthermore, it has been a regular practice in AFib detection to divide EKG recordings into fixed-length subsets [24,30,38]. Pham et al. used EKG segments of both 2 s and 5 s and compared the performance of the two segmenting methods [38]. Liaqat et al. solely incorporated 10-s subset lengths [30] for AFib detection. The 2020 study on Lorenz plots for AFib detection utilized a subset length of 60 R-R intervals [14]. Additionally, a study on entropy-based AFib detection divided the recordings into segments of 30 beats [39]. However, based on our knowledge, there have been no previous studies investigating the impact of different segmentation lengths on classification performance measures, which is one of the goals of this study.

### 1.3. Contribution

The Markov transition model (transition matrix), which classifies R-R intervals based on a running mean and records the proportion of R-R interval transitions [17], is a robust measure of R-R interval variability that has not yet been utilized in conjunction with machine learning for AFib detection. To evaluate the effectiveness of the transition matrix features, we will consider both tree-based and permutation feature importance measures. The permutation importance measure, which uses the decrease in model accuracy to calculate the importance of individual features, has not been previously used for AFib detection [40]. For a given segmentation scheme, the most significant features for each classifier could be incorporated into an algorithm that runs on a wearable device.

The three primary contributions of this study are outlined below:Feeding a total of 21 features, including 8 innovative transition matrix features, into 11 machine learning classifiers, and systematically comparing the classifier outputs to determine the most effective models for AFib detection.Using two feature importance methods, including the permutation feature importance and tree-based feature importance, to obtain the most significant and predictive features in AFib detection.Incorporating three segmentation schemes of 5 s, 10 s, and 25 s to examine the effect of different segment lengths on classifier performance and feature importance results.

Figure 2 provides a flowchart for the overall proposed AFib detection framework.

The remainder of this paper is organized as follows. Section 2 explains the exploratory data analysis and preprocessing performed on the MIT-BIH dataset. In Section 3, the 21 features considered in this research are introduced. Section 4 describes all 11 machine learning classifiers used. In Section 5, the feature importance methods are explained. Section 6 provides the experiment results, including the model performance and feature importance results. Section 7 discusses the principle findings and outlines areas of future work. In Section 8, the main conclusions are summarized.

## 2. Exploratory Data Analysis and Preprocessing on MIT-BIH

### 2.1. MIT-BIH Atrial Fibrillation Database

The MIT-BIH Atrial Fibrillation database [41] contains two-signal electrocardiogram (EKG) recordings of 25 human subjects, 23 of which were complete and usable. The EKG recordings were sampled using a professional ambulatory EKG device, resulting in minimal signal noise. Each 10-h recording includes professional annotations indicating the chronological location and rhythm type of individual heartbeats. The rhythms were marked as either Atrial Fibrillation (AF), Atrial Flutter (AFL), Junctional Rhythm (J), or Normal (N). Two EKG nodes recorded at a rate of 250 Hz and registered upwards of 9 million individual signals each, totaling over 18 million signals per subject. The location of the R peaks for each beat and the labels for each rhythm were obtained. Once the R peaks were determined, the R-R intervals were calculated as the difference between two consecutive R peaks. Since the goal of this research involves detecting AFib from other rhythms, the AFL, J, and N rhythms were all grouped together as “Non-AFib”, and each heartbeat was marked as “AFib” or “Non-AFib” based on the label of its corresponding rhythm.

### 2.2. Single-Subject Analysis

Single-subject analysis, the removal of extreme outliers, and rhythm differentiation were all part of data analysis and processing on the MIT-BIH dataset. Exploratory data analysis (EDA) was first performed on all subjects to visualize the distribution of their R-R intervals. A line plot and histogram of the R-R interval distribution for Subject 04015 is depicted in Figure 3. The line plot shows that the R-R intervals for Subject 04015 had a baseline around or slightly greater than 0.8 s, with regular fluctuations from this baseline. The histogram of the R-R intervals further illustrates that the most frequent R-R interval size was around 0.8 s. The continuous line provides the kernel density estimate for the distribution of the R-R intervals. In our preliminary studies, similar graphs were created for all the other subjects to gain a strong understanding of the R-R interval distribution for each subject. Results indicated that almost all of the subject’s R-R intervals were located in the range of 0.2 s to 1.6 s.

After performing single-subject analysis, the R-R intervals of all 23 subjects were collectively graphed and compared, as shown in Figure 4. The horizontal black bars depict the mean R-R interval length for each subject.

### 2.3. Outlier Removal

Based on Figure 4, it was apparent that the R-R intervals of many subjects contained extreme outliers. These outliers were most likely caused by noise, and were, thus, filtered from the data. The filter cutoff point was set at an R-R interval length of 2 s. Since the instantaneous heart rate can be calculated by dividing 60 by the R-R interval length, an R-R interval of 2 s results in a heart rate of 30 beats per minute (BPM). R-R intervals longer than 2 s correspond to a heart rate below 30 BPM, a lethally low level. Therefore, any R-R intervals longer that 2 s were filtered from the data.

### 2.4. Rhythm Differentiation

Upon filtering for outliers, we differentiated and located the different types of rhythms within each EKG to better understand which sections of R-R intervals corresponded to certain types of heartbeat rhythms. The pie charts in Figure 5 depict the proportion of Normal, AFib, and Other rhythms for Subjects 04936 and 07910, respectively. Subject 04936 has a significantly higher proportion of AFib rhythms (73.9%) than normal rhythms (18.8%). Additionally, 7.3% of this subject’s heart beats are labeled as “Other”. On the other hand, Subject 07910 has the highest proportion of normal heart beats (84.5%) and a very small proportion of rhythms labeled as “Other” (1.4%).

Figure 6 illustrates the location of Normal, AFib, and Other rhythms for Subject 04015 and Subject 04043. Subject 04015 has very few AFib episodes and is primarily experiencing Normal rhythms. The few AFib rhythms of this subject correspond to R-R interval sizes that deviate from the baseline. In contrast, Subject 04043 is undergoing a significantly greater proportion of AFib episodes. As expected, the rhythms labeled “AFib” are clustered in areas with higher R-R interval variability, while the rhythms around the baseline of 0.55 to 0.65 ss are primarily labeled as “Normal”.

Finally, all EKG recordings were divided using three segmentation schemes: 5 s, 10 s, and 25 s. The 5-s [38] and 10-s [30] segmentation schemes were used based on previous studies. The 25-s segments were also included to test the effect of a larger subset length. Feature extraction and classification were performed independently for each subset length. A subset was classified as “AFib” or “Non-AFib” based on whether or not the majority of the rhythms in that subset were labeled as “AFib”.

## 3. Features

We considered a total of 21 R-R interval derived features in this study, described below.

### 3.1. Transition Matrix Features

The transition matrix features, established by Moody and Mark [17], involve classifying each R-R interval as short, medium, or long, based on how it compares to the running mean. The formula for the running mean is provided below, where $RR_{i}$ is the length of the *i*th R-R interval, $rmean_{i}$ is the current running mean, and $rmean_{i-1}$ is the previous running mean.
(1)$$rmean_{i}=0.75\times rmean_{i-1}+0.25\times RR_{i}.$$

An R-R interval was classified as “short” if it was less than 85% of the running mean and “long” if it was greater than 115% of the running mean. All other R-R intervals were classified as “medium”. The transitions between R-R intervals were labeled based on the classifications of the two R-R intervals comprising that transition. For example, a transition from a short R-R interval to a regular R-R interval would be labeled as a short-to-regular (StoR) transition. Table 1 shows the nine possible transition types, each of which was considered to be an individual transition matrix feature.

The proportional presence of the nine transitions between R-R intervals for Subject 04936 is illustrated in Figure 7. As expected, the regular-to-regular (RtoR) transitions type is most prevalent. However, since this subject is experiencing a majority of AFib rhythms, as seen in Figure 5, the other eight transition types are also present and collectively comprise almost 50% of the transitions.

Figure 8 illustrates the nine transition frequencies for AFib and non-AFib subsets. For the majority of Non-AFib subsets, the RtoR transitions type is the most common, comprising over 90% of the transitions. The other eight transition types occur very infrequently, often around 0%. The distribution of transition proportions is more balanced for AFib subsets. The RtoR transitions frequency is significantly less dominant than in the Non-AFib subsets, comprising less than 60% of the transitions for the majority of AFib subsets. The remaining eight transitions were more prevalent than in the Non-AFib subsets.

Since the summation of all nine transition percentages is one, using all features would introduce collinearity into the model. Thus, when feeding the transition matrix features into the models, only eight of the nine transition types were utilized. Because the RtoR transitions type is present in higher percentages than the other transition matrix features, to maintain the diversity of the features, the RtoR feature was dropped when training and testing the machine learning models.

### 3.2. RR Variance

The RR Variance ($RR_{var}$) feature, developed by Ghodrati et al., involves finding the absolute difference between two consecutive R-R intervals and dividing it by the running mean [26]. Ghodrati et al. calculated the running mean by multiplying the previous running mean by 0.9 and adding it to the product of 0.1 and the length of the current R-R interval [26]. For the purposes of consistency, the running mean, as shown in Equation (Equation 1), was utilized as for the transition matrix features.
$$RRvar_{i}=\frac{|RR_{i}-RR_{i-1}|}{rmean_{i}}.$$

Since $RR_{var}$ is a measure of R-R interval variability, subsets with higher $RR_{var}$ values are more likely to be associated with AFib episodes.

### 3.3. $RR_{200}$

The $RR_{200}$ feature is a variation of the $RR_{\left(100\right)}$ algorithm, which was proposed by Young et al. [25]. The $RR_{100}$ algorithm involves creating a count variable, increasing the count by 1 if the difference between two consecutive RR intervals is greater than 100 ms and decreasing the count by 1 if the difference between two R-R intervals is less than 100 ms. If the value of count is greater than 6, the subset is classified as AFib [25].

Instead of using a cutoff of 100 ms, our preliminary studies in [42] suggested that a cutoff of 200 ms was more optimal for AFib detection. Thus, our feature utilized a cutoff of 200 ms to determine whether to increase or decrease the count by 1, as shown by the equations below. The count was divided by the sum of the length of the R-R intervals in the subset to yield the $RR_{200}$ feature.
$$RR_{i}-RR_{i-1}>200\;\mathrm{ms}\overset{}{\to }\mathrm{count}\;=\;\mathrm{count}\;+\;1,$$
$$RR_{i}-RR_{i-1}<200\;\mathrm{ms}\overset{}{\to }\mathrm{count}\;=\;\mathrm{count}\;-\;1,$$
$$RR_{200}=\frac{\mathrm{count}}{\sum RR_{i}}.$$

A higher value for $RR_{200}$ would indicate that the differences between consecutive R-R intervals are more extreme, and thus, correspond to a higher probability of AFib.

### 3.4. Root Mean Square of the Successive Differences

The root mean square of the successive differences (RMS) measures the average difference, or variability, between two consecutive R-R intervals. It is calculated using the following formula, where *n* is the number of R-R intervals in the subset [28]:$$RMS=\sqrt{\frac{1}{n-1}\sum_{i=1}^{n-1}{\left(RR_{i+1}-RR_{i}\right)}^{2}}.$$

### 3.5. Standard Deviation

The standard deviation ($\sigma_{RR}$) is defined as follows:$$\sigma_{RR}=\sqrt{\frac{1}{n-1}\sum_{i=1}^{n}{\left(RR_{i}-\mu_{RR}\right)}^{2}},$$
where $\mu_{RR}$ is the mean R-R interval length in a subset.

### 3.6. Median Absolute Deviation (MAD)

The Median Absolute Deviation (MAD), as proposed by Kennedy [28], measures the median distance of each data point from the median. The MAD is a more robust feature than the standard deviation because it does not succumb easily to outliers and works well with non-normal distributions.
$$MAD=median\left(|RR_{i}-\tilde{RR}|\right),$$
where $\tilde{RR}$ is the median R-R interval length in a subset.

### 3.7. Coefficient of Variance (CoefVar)

The Coefficient of Variance (CoefVar) is the standard deviation of the R-R Intervals divided by the mean of the R-R intervals [28]. It is a simplified measure of variance compared to standard deviation since it outputs lower numbers:$$CoefVar=\frac{\sigma_{RR}}{\mu_{RR}}.$$

### 3.8. Interquartile Range and Range

The interquartile range and range are used as measures of variability and were adopted in this study:IQR=Q3−Q1,
Range=Max−Min.

### 3.9. Gini Index

The Gini Index is the area between the Lorenz Curve and the line of perfect equality. It is used as a quantitative measure of inequality among values in a population [43]. In the case of AFib detection, the Gini Index measures the inequality, or variation, between R-R intervals. The Gini Index for each subset was calculated using the following formula:$$G=\frac{2}{n}\frac{\sum_{i=1}^{n}i\times RR_{i}}{\sum_{i=1}^{n}RR_{i}}-\frac{n+1}{n}.$$

### 3.10. Poincare Plot Distance

The Poincare Plot graphs each R-R interval against its following R-R interval in the rhythm. The Poincare plot of an AFib positive subset should show no pattern or correlation. The following feature, proposed in [44], was used to assess the variability of the R-R intervals in a Poincare plot:$$PP=\frac{1}{n-2}\sum_{i=1}^{n-2}\sqrt{{\left(RR_{i}-RR_{i+1}\right)}^{2}+{\left(RR_{i+1}-RR_{i+2}\right)}^{2}}.$$
This feature calculates average distance between two consecutive points in the Poincare plot.

### 3.11. Entropy Features

Three different measures of entropy were also used as features: Approximate Entropy, Sample Entropy, and Shannon Entropy.

#### 3.11.1. Approximate Entropy

Approximate entropy is an excellent measure of unpredictability in time-series data [45,46]. Approximate entropy is calculated with the following formula:$$ApEn=\phi^{m}\left(r\right)-\phi^{m+1}\left(r\right),$$
$$\phi^{m}\left(r\right)=\frac{1}{n-m+1}\sum_{i=1}^{n-m+1}\log\left(C_{i}^{m}\left(r\right)\right),$$
where *n* represents the number of R-R intervals in the subset, *m* is the length of the pattern, and *r* is the tolerance. In our study, for every subset, *m* was set to 2, while *r* was 0.2 times the standard deviation of the subset. The tolerance determines the extent to which two datasets can be different. The expression $C_{i}^{m}\left(r\right)$ is determined by counting the number of patterns of length *m* that have a distance less than *r* and dividing this count by n−m+1.

#### 3.11.2. Sample Entropy

Sample Entropy is a modification of the approximate entropy and is calculated by the following formula [45,46]:$$SampEn=-\log(\frac{A}{B}),$$
$$A=\sum_{i=1}^{n-m}\left(C_{i}^{m+1}\left(r\right)-1\right),$$
$$B=\sum_{i=1}^{n-m+1}\left(C_{i}^{m}\left(r\right)-1\right).$$
where $C_{i}^{m}\left(r\right)$ is the same as for approximate entropy. The values of *m* and *r* are also the same as those used for approximate entropy. The expression $\frac{A}{B}$ is the probability that if two datasets of length *m* have a distance less than *r* then two datasets of length m+1 will also have a distance less than *r*. The Sample Entropy is the negative logarithm of this probability.

#### 3.11.3. Shannon Entropy

We propose the combination of the Shannon Entropy feature with the transition matrix proportions. The Shannon Entropy measures the uncertainty of a probability distribution [46]. In the context of AFib detection, the Shannon Entropy measures the uncertainty in the R-R intervals. The Shannon entropy (H) for each subset was computed using the following formula:$$H=-\sum_{i=1}^{n}p_{i}\log_{2}p_{i}.$$
where $p_{i}$ represented the proportion of a particular type of transition between two R-R intervals (StoS, StoR, etc.). Since there are nine possible transitions, each subset had at most nine different $p_{i}$ values. For each proportion, the expression $p_{i}\log_{2}p_{i}$ was evaluated and the values for this expression were summed together to receive the Shannon Entropy. The greater the Shannon Entropy, the higher the variability of the R-R intervals, indicating a higher chance of AFib.

### 3.12. Features across AFib and Non-AFib Classes

Figure 9 provides a comparison of the MAD, Sample Entropy, R-R variance, and Coefficient of Variance (CoefVar) in AFib and non-AFib classes. The MAD, R-R variance, and Coefficient of Variance are significantly higher for AFib compared to non-AFib. The Sample Entropy is generally higher for AFib as well. Therefore, greater values for these features can be used as indicators of atrial fibrillation. In our preliminary studies, similar boxplots were created for all features, and it was observed that every feature was greater for AFib than non-AFib.

## 4. Classifiers

The following classification models were used on the MIT-BIH dataset: Non-tree based classifiers, including Logistic Regression [47], K-Nearest Neighbors (KNN) [48], Linear Discriminant Analysis (LDA) [49], and Quadratic Discriminant Analysis (QDA) [50], as well as tree-based classifiers, including Decision Trees [51], Bagging [52], Random Forest [53], Adaptive Boosting [54], Gradient Boosting [55], Light Gradient Boosting [56], and Extreme Gradient Boosting [57].

### 4.1. Non-Tree Based Classifiers

Logistic Regression, Linear Discriminant Analysis (LDA), Quadratic Discriminant Analysis (QDA), and K-Nearest Neighbors (KNN) were the four non-tree based models used for AFib classification. Logistic Regression is a reliable model for binary dependent variable classification and binds the dependent variable so that the value returned is always between 0 and 1 [47]. LDA determines a line of best fit which divides a given set of features into two categories [49]. QDA, similar to LDA, produces an idealized function that classifies groups, but divides a set of data based on non-linear functions. For this reason, it is helpful for more convoluted features. On the other hand, KNN classifies a given point by taking a simple or weighted vote of the *k* nearest points [48]. We utilized KNN-CV for classification, a version of the KNN algorithm that determines the ideal value of *k* using cross validation. Our model used a simple majority vote with no distance-based weighting. The ideal value of *k* differed based on the features used. This classification is highly sensitive to the local topography of a dataset.

### 4.2. Tree-Based Classifiers

Seven tree-based classifiers were considered in this study. The Decision Tree (DT) classifier divides the predictor space into regions to perform classification. These regions are based on the minimization of the Gini Index, a measure that calculates the probability that a data point is incorrectly classified [51]. Decision Trees form the basis for all other tree methods, including Bagging, Random Forest, Adaptive Boosting, and the variants of Gradient Boosting.

Bagging uses a combination of bootstrapping and aggregating. Bootstrapping re-samples a dataset of size *n* to create many different samples of size *n* by taking random samples from the original dataset with replacement [52]. A decision tree is then fitted to each of these individual samples. Each tree outputs a different result, and the final prediction is based on a majority vote of each of these different results, a technique known as aggregating. A drawback to bagging is that the presence of strong predictors in the data can create trees that are highly correlated with each other. Thus, Random Forest decorrelates different decision trees by limiting the dominance of strong predictors through randomly selecting predictors to create trees. Random Forest, similar to Bagging, incorporates bootstrapping and aggregating. However, instead of being fed with all predictors, each tree is fed only a limited number, typically the square root of the total number of predictors [53]. AdaBoost “learns” from its mistakes by placing a higher weight on incorrect classifications and a lower weight on correct classifications [54]. It creates many decision trees, changes the weighting of these trees based on its mistakes, and then takes the weighted average of the results of these trees to create a final prediction.

Three variants of gradient boosting methods were adopted in this study: Gradient Boosting (GBM), Light Gradient Boosting (LGBoost), and Extreme Gradient Boosting (XGBoost). Gradient Boosting, similar to AdaBoost, “learns” from its mistakes by training on its error: the residuals between the true value and the predicted value. It creates many decision trees, forming each tree based on the error of the previous tree, with the goal of minimizing the loss function [55]. XGBoost builds on the gradient boosting framework and utilizes level-wise tree growth in which the trees grow an entire level at a time [57]. It is designed to be more accurate and computationally efficient than Gradient Boosting, making it a strong algorithm when dealing with large datasets. LGBoost, which also uses a gradient boosting framework, uses leaf-wise tree growth, in which the decision trees grow one node at a time [56]. This leaf-wise tree growth further reduces loss, allowing for high accuracy and efficiency.

### 4.3. Hyperparameter Tuning

The grid search algorithm via cross-validation was applied to tune the hyperparameters, with the goal of improving the classification accuracy. For the 5-s, 10-s, and 25-s segmentation schemes, various hyperparameter tuning schemes were individually considered for eight different classifiers, listed below: (1) For KNN-CV, k-values of 1 to 35 were studied; (2) For Decision Tree, a max depth of 1 to 25 was considered; (3) For Random Forest, a number of trees of 10, 50, 100, and 500 was tested, along with a max depth of 3 to 20 and a max number of features of 4 and 5; (4) For AdaBoost, learning rates from 0.1 to 1 with increments of 0.1 were considered; (5) For GBM, learning rates from 0.1 to 1 with increments of 0.1, a number of trees of 50, 100, and 500, and a max depth of 3 to 12 were studied; (6) For LGBoost, a learning rate of 0.1 to 1 with increments of 0.1, a number of trees of 10, 50, 100, and 500, a max depth of 3 to 15, and a number of leaves of 6 to 32 were studied; (7) The considered parameters for XGBoost were similar to those of LGBoost except that a max depth of 1 to 15 was examined with the default number of leaves. The random state was fixed for all models to avoid unexpected experimental variability.

### 4.4. Leave-One-Person-Out Cross Validation

Leave-One-Person Out Cross Validation (LOOCV) [58] was employed for the training and testing of all classifiers. LOOCV was used to avoid information leaking from an individual subject, since regular k-fold Cross Validation may contain data from the same subject in both the training and testing sets. For LOOCV, the data were first split into 23 folds, where each fold contained the subsets for one of the 23 subjects. In total, 23 iterations were carried out in which the model trained on data from 22 subjects and tested on the 1 remaining subject. In each iteration, the model was tested on a different subject. This iterative process of training and testing allowed each model to be tested on all of the data.

### 4.5. Metrics

To quantify the performance of the models, six metrics were used: accuracy, sensitivity, specificity, precision, F1-score, and AUC. The Area Under Curve (AUC) is the area under the Receiving Operating Characteristics (ROC) Curve, which graphs the false positive rate against the true positive rate. The AUC-Score is the ability of a model to distinguish between different categories. In the case of AFib detection, AUC determines the ability of a classifier to differentiate between “AFib” and “Non-AFib” classes.

## 5. Feature Importance

Each of the tree-based classifiers, including three gradient boosting models, contain a default feature importance measure as part of the Python Scikit-learn library. These tree-based feature importances were used to identify strongly predictive features in the output of the classifiers. It is noted that tree-based feature importances were not available for the non-tree classifiers such as Logistic Regression, LDA, QDA, and KNN.

Table 2 provides the tree-based feature importance methods used for each tree classifier. Decision Tree, Bagging, Random Forest, and GBM all used the Gini Importance measure, while LGBoost and XGBoost employed the count-based and gain-based importance measures, respectively.

Additionally, the permutation feature importance measure [40], which can be applied to any classifier, was also used to determine the most significant and predictive features to have a consistent measure across all models. The permutation importance calculation involves changing the order of one feature while keeping the remaining 20 features constant and calculating the effect of this permutation on the accuracy of the model. A larger decrease in model accuracy indicates a higher feature importance. This process was carried out for all 21 features independently, with the order of each feature being permuted 10 times, and the decrease in the model accuracy was used to determine the relative importance of each feature.

## 6. Experiment Results

In this section, 21 features, including the proposed transition matrix features, extracted from 5-s, 10-s, and 25-s subsets on the MIT-BIH dataset were compared systematically on 11 machine learning classifiers using LOOCV. Then, permutation and tree-based feature importances were used to determine the most predictive features for each model.

### 6.1. Overall Model Performance for Three Segmentation Schemes

Table 3 provides the performance metrics of all classifiers when fed with 5-s subsets. When the models were trained with 5-s subsets, XGBoost, LGBoost, and Random Forest had the three highest accuracies, at 92.52%, 92.46%, and 92.43%, respectively. Gradient Boosting trailed Random Forest with an accuracy of 92.33%. The three highest sensitivities were exhibited by Logistic Regression (92.84%), LDA (92.55%), and XGBoost (92.00%). Even though Logistic Regression and LDA yielded the two highest sensitivities, they performed relatively poorly in terms of the other five metrics and were, therefore, not considered to be high-performing classifiers.

Table 4 demonstrates that, for the 10-s segmentation scheme, the three gradient boosting models and Random Forest exhibited high accuracies as well. GBM had the highest accuracy at 94.81%, followed by LGBoost (94.77%) and Random Forest (94.76%). XGBoost had the fourth highest accuracy, at 94.70%. The three highest AUC scores were exhibited by LGBoost, XGBoost, and Random Forest, with scores of 98.49%, 98.46%, and 98.45%, respectively. LGBoost (94.93%) and Random Forest (94.81%) had the two greatest sensitivities.

Table 5 provides the results of all classifiers with the 25-s segmentation scheme. The highest performing model was XGBoost with an accuracy of 96.29%, while LGBoost and Gradient Boosting had the second and third highest accuracies, respectively, at 96.11% and 96.08%. Following these accuracies were AdaBoost (96.03%) and Random Forest (95.55%). AdaBoost and Random Forest tied for the highest AUC score, at 99.16%. Therefore, AdaBoost and Random Forest were the most adept at distinguishing between AFib and non-AFib subsets. Additionally, XGBoost (96.98%), Gradient Boosting (96.88%), and LGBoost (96.86%) had the three highest sensitivities.

Across all three subset lengths, the four non-tree based models (Logistic Regression, LDA, QDA, and KNN-CV) yielded the least satisfactory results in terms of accuracy. Decision Tree, Bagging, and Adaptive Boosting generally outperformed the non-tree based models, while Random Forest and the three gradient boosting models demonstrated the highest accuracies.

### 6.2. Comparison of Model Performance across Three Segmentation Schemes

We will systematically compare the performance of the models across 5-, 10-, and 25-s segmentation schemes.

Figure 10 depicts the accuracies of all eleven classifiers for each of the three segmentation schemes (5, 10, and 25 s). When fed with 5-s subsets, XGBoost, LGBoost, Random Forest, and GBM had the four highest accuracies. On the other hand, the four non-tree based classifiers exhibited the four lowest accuracies. Similar results were obtained for the 10-s and 25-s subsets. With the 10-s subsets, the three gradient boosting models and Random Forest had the four highest accuracies. When fed with 25-s subsets, XGBoost, LGBoost, and GBM yielded the three greatest accuracies, with AdaBoost following in fourth. For every classifier, the highest accuracy occurred with the 25-s subsets while the 5-s subsets resulted in the lowest accuracy, demonstrating that subset length and accuracy have a positive concordance across all models.

Figure 11 illustrates the sensitivites of all classifiers across all three subset lengths. Similar to the accuracy, the sensitivity of every classifier increased as subset length increased.

Finally, Figure 12 depicts the trend in AUC score across different subset lengths for all classifiers. The AUC score and subset length have a clear positive concordance. The higher AUC scores were exhibited by the three gradient boosting models, as well as Random Forest, Bagging, and Adaptive Boosting.

### 6.3. Comparison of Actual Rhythms and XGBoost Classification Performance

Figure 13 depicts the actual and predicted subset classifications for Subject 04043 using XGBoost and a 25-s segmentation scheme. The MAD values for each subset are shown on the vertical axis. As expected, the majority of the subsets with higher MADs were classified as “AFib”, while the subsets with lower MADs were classified as “non-AFib”. The predicted values are primarily in agreement with the true classifications, since the XGBoost model had a 96.4% accuracy with Subject 04043. Similar graphs using XGBoost with a 25-s segmentation scheme were created for the other subjects and features.

### 6.4. Feature Importance Results for Three Segmentation Schemes

Table 6, Table 7 and Table 8 provide the top five most predictive features for every classifier with the 5, 10, and 25-s segmentation schemes, respectively. Across all segmentation schemes, the most predictive features differed in various levels between the tree-based importance and permutation importance [59] for almost every subset-classifier combination. It is an interesting finding that the transition matrix features were generally more predictive with the tree-based importance measures than with the permutation importance measure.

LogReg, LDA, QDA, and KNN-CV did not contain tree-based feature importances and could, therefore, only be analyzed with the permutation importance measure. Additionally, the tree-based importance measures varied between the classifiers, making the permutation feature importance more consistent. Since KNN-CV with the permutation importance measure had high computational intensity for the 5- and 10-s segmentation schemes, we only considered ten classifiers for these two subset lengths, while all eleven classifiers were considered for the 25-s segmentation scheme.

Furthermore, Figure 14 depicts the tree-based importance results for GBM, LGBoost, and XGBoost with 25-s subsets. On the other hand, Figure 15 provides the permutation feature importance results for GBM, LGBoost, and XGBoost using the 25-s segmentation scheme. Each bar represents the median of ten permutations.

When the models were fed with 25-s subsets, MAD was the most significant predictor for GBM by a considerable margin, causing a decrease in accuracy of 0.3, the most significant predictor for LGBoost, and the second most predictive feature for XGBoost. Sample Entropy was found to be the strongest predictor with XGBoost. Sample Entropy, MAD, and $RR_{200}$ were consistently in the top four most predictive features across all three gradient boosting models. These three features were considered equally significant as the decrease in accuracy caused by the removal of one of these features was almost equivalent, at around 0.14.

### 6.5. Feature Prevalence Results in the Top Five Important Features for Three Segmentation Schemes

Figure 16 demonstrates the prevalence of each feature in the top five features for the 5-s, 10-s, and 25-s segmentation schemes using both tree-based and permutation importance measures. RR200 (35), IQR (32), MAD (30), Sample Entropy (28), and Coefficient of Variance (27) were among the most important features for both feature importance measures, with the total frequency of each feature in the top-5 most important features provided in parenthesis. R-R Variance (21), Range (16), and RtoL (16) was the next cohort of the most important features. It is noted that RtoL is the best feature among all eight transition matrix features.

### 6.6. Feature Importance Results for the Proposed Transition Matrix Features

The feature importances for our proposed transition matrix features were studied systematically as below. Table 9 provides the rank of the most significant transitions feature for every model-subset combination using the tree-based importance measures. A lower rank number corresponds to a greater feature importance. Across all models, the significance of the most predictive transitions feature increased as the subset length became greater. With the 25-s segmentation scheme, the RtoL transition matrix feature was the most significant predictor for five of the seven tree-based models. The RtoL feature was the most predictive transition matrix feature for all tree-based models when using the 10-s and 25-s segmentation schemes, with the exception of AdaBoost. However, when using the 5-s segmentation scheme, the StoL transitions type was more prominent across the models than the RtoL transitions feature.

For both GBM and XGBoost, the RtoL transition matrix feature was the most predictive feature using the tree-based feature importance. Therefore, based on the tree-based feature importances, the transition matrix was the most significant predictor of AFib for two gradient boosting models. Additionally, for both models, MAD was the second most predictive feature by a considerable margin. However, for LGBoost, the RtoL transitions feature was only the sixth most predictive covariate, while $RR_{200}$ was determined to be the most significant. This difference in the significance of the transitions matrix between LGBoost and the two other gradient boosting variations can be attributed to the variation in the tree-based feature importance measures. In general, the transition matrix features were more predictive when using the tree-based feature importance than with the permutation feature importances. However, the consistency of the permutation feature importance across all models made it a more reliable and robust measure of comparison.

Moreover, Table 10 provides the best transition matrix feature and its rank for every segmentation-classifier combination using the permutation feature importance. The permutation feature importance results varied across different subset lengths. When the models were trained with 5-s subsets, the regular-to-long transition matrix feature was the 13th most predictive covariate for all three gradient boosting variations. Thus, even the best transition matrix feature played a relatively small role in classification. When the models were fed with 10-s subsets, the regular-to-long transition matrix feature was the 11th, 9th, and 8th most predictive covariate for Gradient Boosting, LGBoost, and XGBoost, respectively. Lastly, when the models were fed with 25-s subsets, the regular-to-long transition matrix feature was the 5th most significant covariate for Gradient Boosting and the 7th most predictive for LGBoost and XGBoost. Therefore, across all three gradient boosting classifiers, the transition matrix features were less predictive with 5-s and 10-s subsets than with 25-s subsets.

For every tree and gradient-boosting model, the rank of the most predictive transition matrix feature increased as the subset-length increased; the transition matrix features carried the greatest significance with the 25-s subsets and the least significance with the 5-s subsets. However, this trend did not occur in non-tree based models such as Logistic Regression and LDA, where the best transition matrix feature was approximately equally predictive across all subset lengths.

In summary, the regular-to-long (RtoL) transition matrix feature was the most predictive transitions feature for the majority of the segmentation-classifier combinations and was primarily prominent among the tree and gradient boosting models. Due to the consistently high performance of the three gradient boosting models across all subset lengths, the feature importances of the gradient boosting models were given the highest consideration. Since the regular-to-long (RtoL) transition matrix feature was consistently the most predictive transitions feature for the gradient boosting models, the RtoL permutation and tree-based feature importance was used to assess the importance of the overall transition matrix.

## 7. Discussion

Accuracy, as a baseline measure, was the primary metric used to evaluate the models. Sensitivity and AUC score were also prioritized when determining classifier performance. During AFib detection, minimizing the number of false negatives should be prioritized over minimizing the number of false positives to prevent AFib episodes from going undetected. Leaving AFib episodes undetected would have life-threatening consequences, and a model that is used in wearable devices must minimize the chance that such an error would occur. As such, sensitivity was valued over specificity. The AUC score of a classifier was also given high consideration when evaluating model performance. The AUC score quantitatively measures the ability of each model to distinguish between AFib and non-AFib episodes, which is the primary purpose of the classifiers.

### 7.1. Principal Finding

Across all segmentation schemes, the three gradient boosting models and Random Forest exhibited the highest performance in terms of accuracy, sensitivity, and AUC score. For the 5-s segmentation scheme, XGBoost performed best in terms of accuracy and sensitivity, while Gradient Boosting had the highest AUC score. When the classifiers were fed with 10-s subsets, the highest accuracy, sensitivity, and AUC score were demonstrated by Gradient Boosting, Random Forest, and XGBoost, respectively. With the 25-s segmentation scheme, XGBoost exhibited the highest accuracy and sensitivity, while Random Forest and Adaptive Boosting tied for the greatest AUC score. In general, AdaBoost, Bagging, and Decision Tree performed marginally worse than Random Forest and the three gradient boosting variations, while the four non-tree based models (Logistic Regression, LDA, QDA, and KNN-CV) exhibited the poorest performance.

For all classifiers, as the subset length increased, the six performance metrics (accuracy, sensitivity, specificity, precision, F1-Score, and AUC score) improved. All eleven classifiers had the lowest accuracies with 5-s subsets, improved when trained with 10-s subsets, and exhibited their highest performance with 25-s subsets. This result is shown in more detail in Figure 10. A positive concordance between subset length and classifier performance occurs because larger subsets provide more data to train the models and consequently lead to better AFib detection performance. An advantage of shorter subset lengths is more rapid AFib detection. The 5-s subsets will have at most a 5-s delay in AFib detection, while the 25-s subsets will have a maximum 25-s delay. Therefore, there is a trade-off between AFib detection speed and detection accuracy. The ideal subset length depends on the relative importance of timeliness and accuracy when detecting AFib. In terms of performance, a 25-s segmentation scheme would be preferable to a 5-s or 10-s segmentation scheme, because it would allow for more accurate detection. However, this decision depends on an individual basis and should be made with the consultation of a medical professional.

Based on the tree-based and permutation feature importance results, across all classifiers, the eight transition matrix features were more predictive with longer subset lengths. When using the permutation feature importance and a 5-s segmentation scheme, the transition matrix features were not in the top five most significant predictors for any classifiers. Even with the tree-based importance, a transition matrix feature appeared in the top five features for only one classifier. The lower importance of the transition matrix features with the 5-s subsets could be attributed to the fact that many 5-s subsets do not contain an adequate number of R-R intervals for robust classification. The transition matrix features require a sufficient number of R-R intervals to have an accurate distribution of the nine transition types in the subset. As a result, they did not have relatively high significance when the models were fed with 5-s subsets.

The 25-s subsets contained a greater number of R-R intervals, allowing for the transition matrix features to be more predictive with these subsets. Using the permutation feature importance and a 25-s segmentation scheme, the RtoL transitions feature was in the top three most predictive features for three of the seven tree-based models and in the top five most predictive features for five tree-based models. When using the tree-based importance measures, the RtoL transitions type was the single most predictive feature for five of the seven tree-based models.

Additionally, the division of the transition matrix into nine distinct features, eight of which were used for training and testing, could have lowered the overall importance of the transition matrix. Individually, each of the eight transition types do not carry a considerable importance. Therefore, when the eight transition types are individually fed into models, they are outperformed by other covariates. However, if all eight transitions were to be conglomerated and considered as one feature, the importance of this single predictor would most likely rise considerably.

### 7.2. Future Work

Future work could be carried out with the testing of our models on datasets more representative of real-world raw EKG data, such as the 2017 PhysioNet Challenge Dataset [60]. Since the models were only trained on the MIT-BIH database, which was obtained by ambulatory EKG recordings and contains minimal noise, real-world data with more noise may present additional challenges. As aforementioned, the eight transition matrix features could be combined together and considered as one feature, using grouped feature importances [61]. This would allow for the cumulative importance of the transition matrix to be evaluated, potentially resulting in a higher importance than those of the individual transitions. Moreover, ensemble models, which aggregate the outputs of multiple classifiers to create a final prediction, could be tested. This may allow for higher accuracies of AFib detection.

## 8. Conclusions

This research aims to incorporate the transition matrix as a novel measure of R-R interval variability with other popular features, while combining three segmentation schemes and two feature importance measures to systematically analyze the significance of individual features. As the subset length increased, the performance of the classifiers improved, since the classifiers exhibited the highest performance across all six metrics with the 25-s segmentation scheme and the lowest performance with the 5-s segmentation scheme. Across all three subset lengths, Random Forest, GBM, LGBoost, and XGBoost were the four highest performing models in terms of accuracy, sensitivity, and AUC-score. The three highest accuracies were demonstrated by XGBoost, LGBoost, and GBM, at 96.29%, 96.11%, and 96.08%, respectively, when fed with 25-s subsets. Additionally, the importance of the transition matrix features substantially increased as the subset length became greater. For both permutation and tree-based feature importance measures, the RtoL transition matrix feature was consistently among the most predictive features with the 25-s segmentation scheme. Overall, the highest performance was exhibited by the gradient boosting models with the 25-s segmentation scheme, while the transition matrix features became increasingly important as the subset length increased.

## Figures and Tables

**Figure 1 sensors-23-03700-f001:**
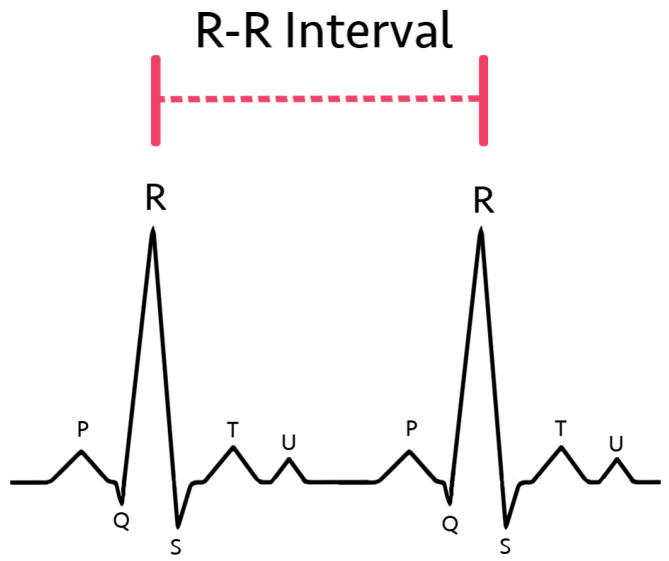
R-R interval diagram in the classical EKG curve.

**Figure 2 sensors-23-03700-f002:**
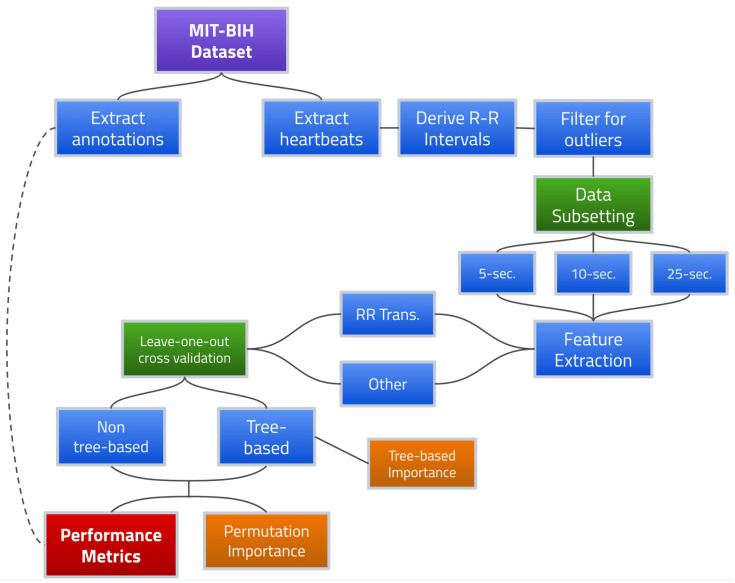
AFib detection framework using R-R interval transition matrix features.

**Figure 3 sensors-23-03700-f003:**
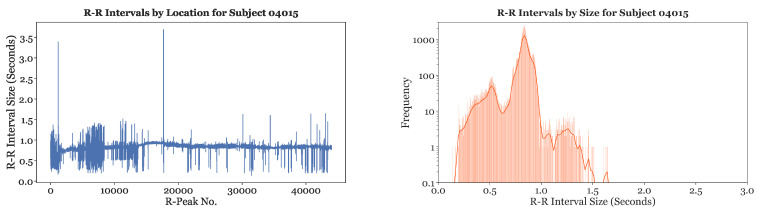
Sample of the line plot and histogram of R-R intervals for Subject 04015.

**Figure 4 sensors-23-03700-f004:**
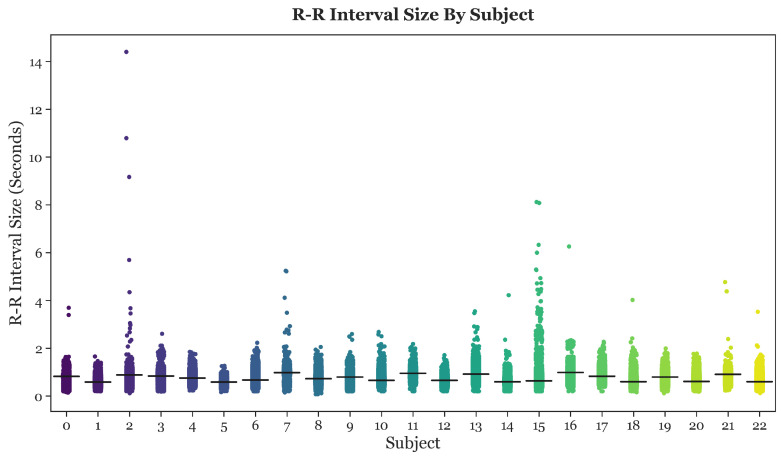
Strip plots of the R-R intervals for all 23 subjects.

**Figure 5 sensors-23-03700-f005:**
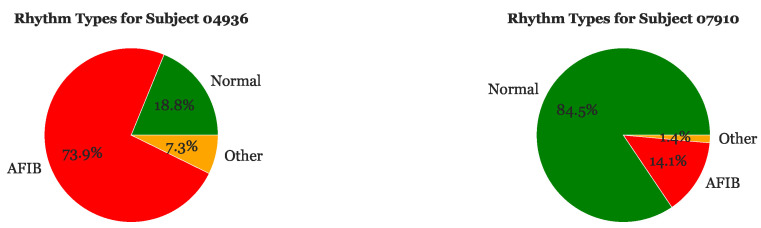
Sample of rhythm frequencies for Subject 04936 and Subject 07910.

**Figure 6 sensors-23-03700-f006:**
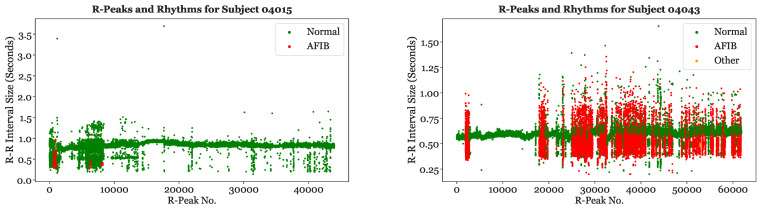
Labeled rhythms for Subject 04015 and Subject 04043.

**Figure 7 sensors-23-03700-f007:**
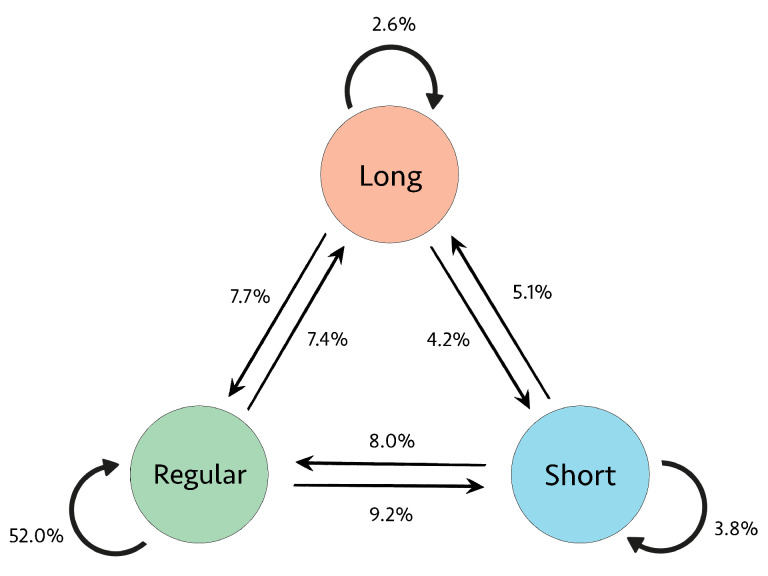
Transition proportions for Subject 04936.

**Figure 8 sensors-23-03700-f008:**
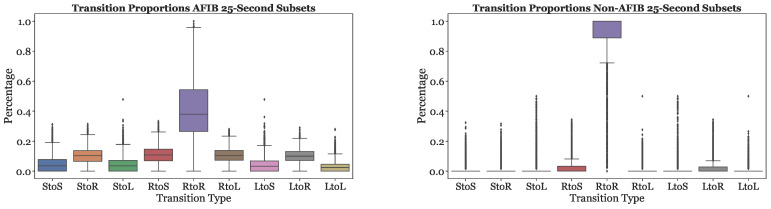
Transition frequencies across AFib and Non-AFib 25-s subsets for all subjects.

**Figure 9 sensors-23-03700-f009:**
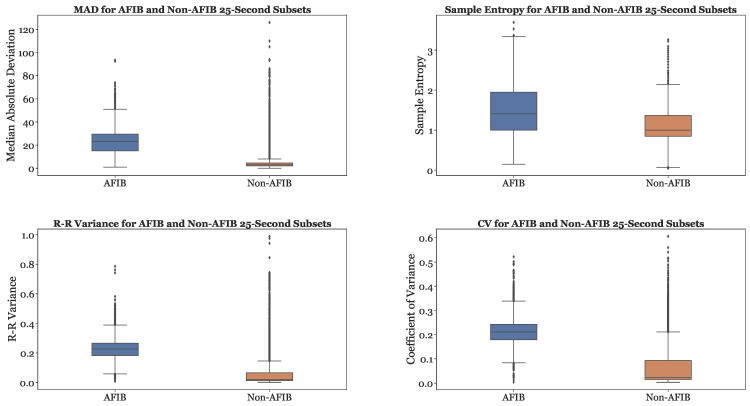
Sample features of MAD, Sample Entropy, R-R Variance, and Coefficient of Variance (CoefVar) Across AFib and Non-AFib Subsets.

**Figure 10 sensors-23-03700-f010:**
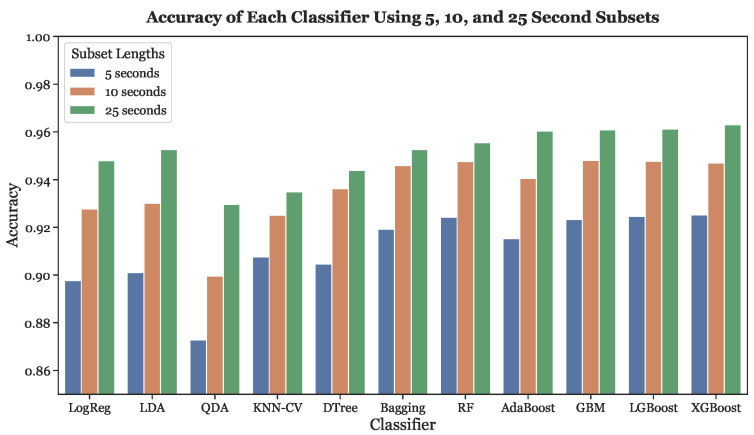
Accuracies across all subset lengths (5, 10, and 25 s) using LOOCV.

**Figure 11 sensors-23-03700-f011:**
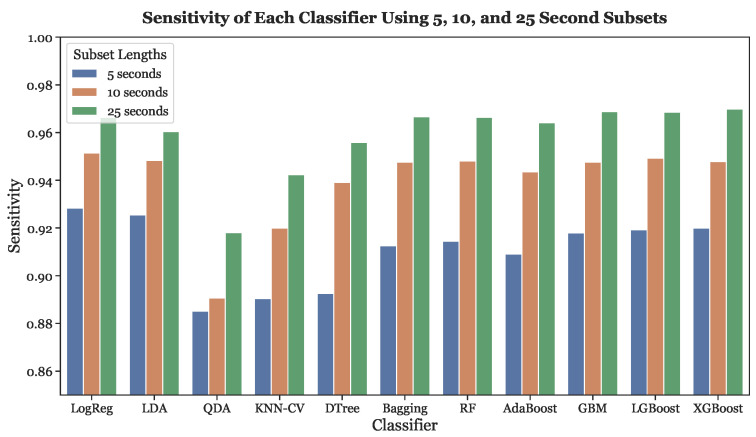
Sensitivities across all subset lengths (5, 10, and 25 s) using LOOCV.

**Figure 12 sensors-23-03700-f012:**
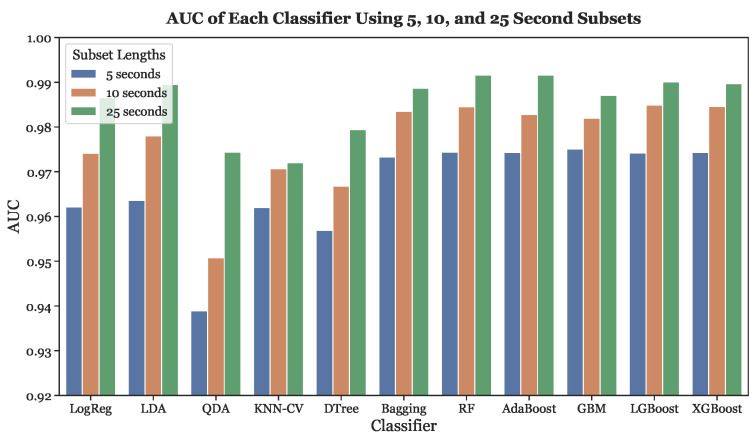
AUC across all subset lengths (5, 10, and 25 s) using LOOCV.

**Figure 13 sensors-23-03700-f013:**
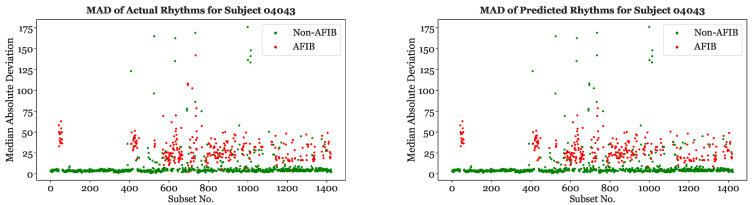
Actual (**left**) and XGBoost based (**right**) rhythms on 25-s subsets for Subject 04043.

**Figure 14 sensors-23-03700-f014:**
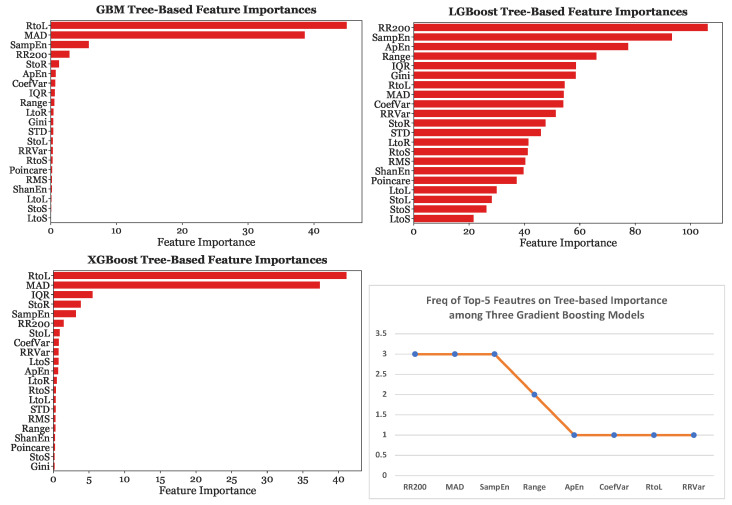
Tree-based importance for Gradient Boosting Models with the 25-s segmentation scheme.

**Figure 15 sensors-23-03700-f015:**
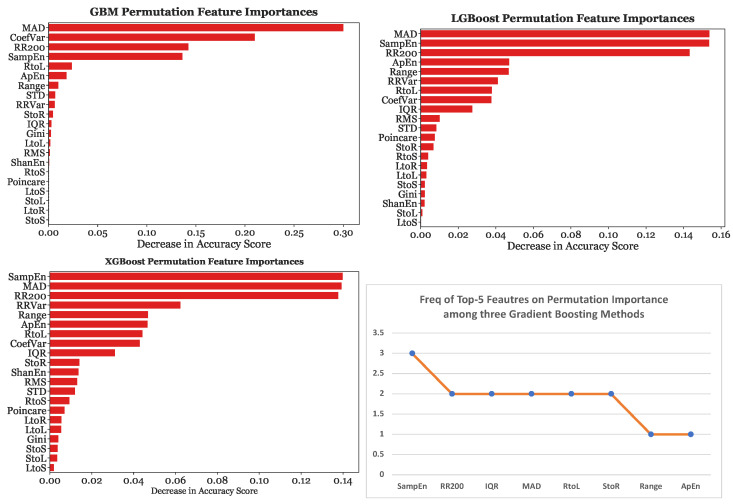
Permutation importances for gradient boosting models with the 25-s segmentation scheme.

**Figure 16 sensors-23-03700-f016:**
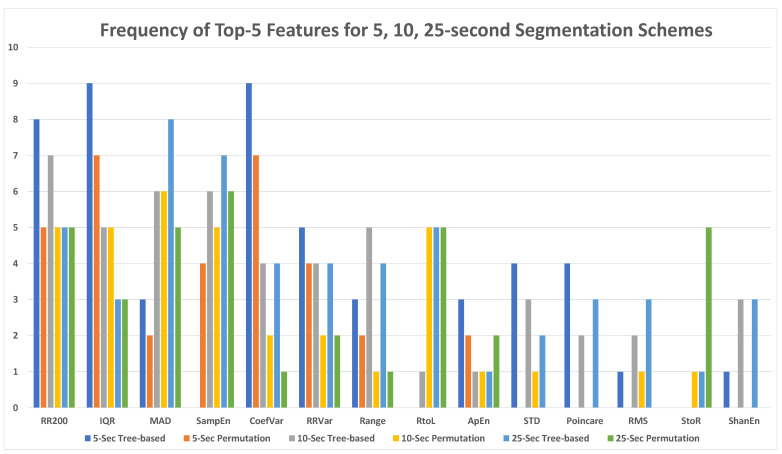
Top five features for all models with 5-, 10-, and 25-s segmentation schemes.

**Table 1 sensors-23-03700-t001:** Illustration of the R-R interval transition matrix.

	To:			
From:		Short	Regular	Long
	Short	StoS	StoR	StoL
	Regular	RtoS	RtoR	RtoL
	Long	LtoS	LtoR	LtoL

**Table 2 sensors-23-03700-t002:** Feature importance measures for tree-based models.

Tree Classifier	Feature Importance Measure
Decision Tree	Gini Importance
Bagging	Gini Importance
RandomForest	Gini Importance
AdaBoosting	Gini Importance
Gradient Boosting	Gini Importance
LGBoost	Count-Based Importance
XGBoost	Gain-Based Importance

**Table 3 sensors-23-03700-t003:** Results for all models with 5-s segmentation scheme using LOOCV. Top three results are highlighted for each measurement.

Model	Accuracy	Sensitivity	Specificity	Precision	F1-Score	AUC Score
LogReg	0.8977	**0.9284**	0.8502	0.9018	0.9149	0.9621
LDA	0.9010	**0.9255**	0.8628	0.9090	0.9172	0.9636
QDA	0.8728	0.8852	0.8526	0.8989	0.8920	0.9389
KNN-CV	0.9076	0.8904	**0.9317**	0.9507	0.9196	0.9620
Decision Tree	0.9047	0.8926	0.9208	0.9435	0.9174	0.9569
Bagging	0.9192	0.9125	0.9277	0.9492	0.9305	0.9733
Random Forest	**0.9243**	0.9145	**0.9375**	**0.9559**	**0.9347**	**0.9744**
AdaBoost	0.9153	0.9091	0.9230	0.9459	0.9271	**0.9743**
GBM	0.9233	0.9179	0.9299	0.9509	0.9341	**0.9751**
LGBoost	**0.9246**	0.9193	0.9311	**0.9519**	**0.9353**	0.9742
XGBoost	**0.9252**	**0.9200**	**0.9315**	**0.9521**	**0.9358**	**0.9743**

**Table 4 sensors-23-03700-t004:** Results for all models with 10-s segmentation scheme using LOOCV. Top three results are highlighted for each measurement.

Model	Accuracy	Sensitivity	Specificity	Precision	F1-Score	AUC Score
LogReg	0.9277	**0.9514**	0.8910	0.9284	0.9398	0.9741
LDA	0.9301	**0.9483**	0.9015	0.9347	0.9415	0.9780
QDA	0.8995	0.8906	0.9114	0.9373	0.9133	0.9508
KNN-CV	0.9250	0.9200	0.9312	0.9521	0.9358	0.9707
Decision Tree	0.9362	0.9391	0.9306	0.9527	0.9458	0.9668
Bagging	0.9459	0.9476	0.9423	0.9607	0.9541	0.9835
Random Forest	**0.9476**	0.9481	**0.9459**	**0.9631**	**0.9555**	**0.9845**
AdaBoost	0.9405	0.9435	0.9349	0.9557	0.9496	0.9828
GBM	**0.9481**	0.9476	**0.9479**	**0.9644**	**0.9559**	0.9820
LGBoost	**0.9477**	**0.9493**	0.9445	0.9621	**0.9557**	**0.9849**
XGBoost	0.9470	0.9479	**0.9446**	**0.9622**	0.9550	**0.9846**

**Table 5 sensors-23-03700-t005:** Results for all models with the 25-s segmentation scheme using LOOCV. Top three results are highlighted for each measurement.

Model	Accuracy	Sensitivity	Specificity	Precision	F1-Score	AUC Score
LogReg	0.9479	0.9664	0.9195	0.9471	0.9567	0.9866
LDA	0.9526	0.9604	0.9402	0.9599	0.9602	0.9895
QDA	0.9296	0.9180	0.9460	0.9620	0.9395	0.9744
KNN-CV	0.9349	0.9423	0.9229	0.9480	0.9452	0.9720
Decision Tree	0.9438	0.9559	0.9249	0.9500	0.9529	0.9794
Bagging	0.9526	0.9666	0.9310	0.9543	0.9604	0.9887
Random Forest	0.9555	0.9664	0.9384	0.9590	0.9627	**0.9916**
AdaBoost	0.9603	0.9641	**0.9540**	**0.9690**	0.9666	**0.9916**
GBM	**0.9608**	**0.9688**	0.9485	0.9656	**0.9672**	0.9871
LGBoost	**0.9611**	**0.9686**	**0.9495**	**0.9662**	**0.9674**	**0.9901**
XGBoost	**0.9629**	**0.9698**	**0.9522**	**0.9680**	**0.9689**	0.9897

**Table 6 sensors-23-03700-t006:** Top five features for all models with the 5-s segmentation scheme.

Model	Tree-Based Importance Top 5 Features	Permutation Importance Top 5 Features
LogReg	N/A	Range, CoefVar, IQR, STD, Poincare
LDA	N/A	Range, ShanEn, IQR, STD, $RR_{200}$
QDA	N/A	STD, RMS, CoefVar, Range, RRVar
Decision Tree	IQR, CoefVar, $RR_{200}$, SampEn, ApEn	CoefVar, IQR, STD, $RR_{200}$, ApEn
Bagging	IQR, CoefVar, $RR_{200}$, SampEn, ApEn	CoefVar, IQR, $RR_{200}$, ApEn, MAD
Random Forest	IQR, MAD, CoefVar, RRVar, Range	IQR, MAD, $RR_{200}$, ApEn, CoefVar
AdaBoost	CoefVar, $RR_{200}$, IQR, Range, RRVar	CoefVar, $RR_{200}$, RRVar, IQR, MAD
GBM	IQR, MAD, $RR_{200}$, CoefVar, SampEn	RRVar, IQR, Poincare, RMS, CoefVar
LGBoost	$RR_{200}$, Gini, IQR, CoefVar, RRVar	IQR, RRVar, Poincare, CoefVar, RMS
XGBoost	IQR, SampEn, StoL, CoefVar, RRVar	IQR, CoefVar, RRVar, Poincare, $RR_{200}$

**Table 7 sensors-23-03700-t007:** Top five features for all models with the 10-s segmentation scheme.

Model	Tree-Based Importance Top 5 Features	Permutation Importance Top 5 Features
LogReg	N/A	Range, ShanEn, Poincare, CoefVar, STD
LDA	N/A	ShanEn, Poincare, Range, RMS, STD
QDA	N/A	STD, RMS, RRVar, CoefVar, ShanEn
Decision Tree	IQR, MAD, $RR_{200}$, SampEn, RtoL	IQR, CoefVar, $RR_{200}$, MAD, RtoL
Bagging	IQR, MAD, $RR_{200}$, SampEn, RtoL	IQR, $RR_{200}$, SampEn, RtoL, MAD
Random Forest	MAD, IQR, RRVar, CoefVar, RtoL	MAD, IQR, $RR_{200}$, SampEn, ApEn
AdaBoost	CoefVar, $RR_{200}$, RMS, RRVar, STD	CoefVar, RRVar, $RR_{200}$, SampEn, Range
GBM	IQR, MAD, $RR_{200}$, SampEn, RtoL	$RR_{200}$, SampEn, IQR, MAD, Range
LGBoost	$RR_{200}$, SampEn, MAD, ApEn, Range	RRVar, MAD, $RR_{200}$, IQR, SampEn
XGBoost	RtoL, MAD, IQR, StoR, SampEn	SampEn, MAD, $RR_{200}$, RRVar, Range

**Table 8 sensors-23-03700-t008:** Top five features for all models with the 25-s segmentation scheme.

Model	Tree-Based Importance Top 5 Features	Permutation Importance Top 5 Features
Logistic Regression	N/A	ShanEn, Range, Poincare, STD, RMS
LDA	N/A	ShanEn, Poincare, CoefVar, STD, RMS
QDA	N/A	RtoS, RRVar, ShanEn, StoR, LtoR
KNN-CV	N/A	IQR, Range, Poincare, RMS, MAD
Decision Tree	RtoL, MAD, SampEn, $RR_{200}$, StoR	MAD, RtoL, IQR, SampEn, $RR_{200}$
Bagging	RtoL, MAD, SampEn, $RR_{200}$, StoR	RRVar, MAD, CoefVar, RtoL, SampEn
Random Forest	RtoL, MAD, IQR, RRVar, StoR	MAD, SampEn, RtoL, $RR_{200}$, IQR
AdaBoost	SampEn, $RR_{200}$, CoefVar, ApEn, RRVar	RRVar, SampEn, RtoL, CoefVar, MAD
GBM	RtoL, MAD, SampEn, $RR_{200}$, StoR	MAD, CoefVar, $RR_{200}$, SampEn, RtoL
LGBoost	$RR_{200}$, SampEn, ApEn, Range, IQR	MAD, SampEn, $RR_{200}$, ApEn, Range
XGBoost	RtoL, MAD, IQR, StoR, SampEn	SampEn, MAD, $RR_{200}$, RRVar, Range

**Table 9 sensors-23-03700-t009:** Best transition matrix feature rank using tree-based importance.

Model	5-s	10-s	25-s
Decision Tree	6 (StoL)	5 (RtoL)	1 (RtoL)
Bagging	6 (StoL)	5 (RtoL)	1 (RtoL)
Random Forest	12 (RtoL)	5 (RtoL)	1 (RtoL)
AdaBoost	12 (RtoS)	13 (RtoS)	9 (LtoR)
GBM	7 (StoL)	5 (RtoL)	1 (RtoL)
LGBoost	13 (RtoL)	9 (RtoL)	7 (RtoL)
XGBoost	3 (StoL)	3 (RtoL)	1 (RtoL)

**Table 10 sensors-23-03700-t010:** Best transition matrix feature rank using permutation feature importances.

Model	5-s	10-s	25-s
LogReg	8 (StoL)	10 (StoL)	8 (StoL)
LDA	6 (StoL)	6 (StoL)	6 (StoL)
QDA	8 (RtoS)	8 (RtoS)	1 (RtoS)
Decision Tree	6 (StoL)	5 (RtoL)	2 (RtoL)
Bagging	7 (StoL)	4 (RtoL)	5 (RtoL)
Random Forest	8 (StoL)	6 (RtoL)	3 (RtoL)
AdaBoost	10 (RtoL)	10 (RtoL)	3 (RtoL)
GBM	13 (RtoL)	11 (RtoL)	5 (RtoL)
LGBoost	13 (RtoL)	9 (RtoL)	7 (RtoL)
XGBoost	13 (RtoL)	8 (RtoL)	7 (RtoL)

## Data Availability

The publicly available MIT-BIH Atrial Fibrillation Database was analyzed in this study. The data can be found here: https://physionet.org/content/afdb/1.0.0/. Access date: 1 May 2021.

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
