# Peer review of "A Study of R-R Interval Transition Matrix Features for Machine Learning Algorithms in AFib Detection"

_sensors, 2023, doi:10.3390/s23073700_

Round 1
Reviewer 1 Report
The paper suggests a method for AFib detection based on feature engineering. Several classic ML models are applied to the engineered features, and experimental evaluation is performed on the MIT-BIH dataset with different time segment lengths.
General comments:
The authors need to compare their suggested methods to modern neural methods that report better scores.
A paper can be better organized, and a proper ablation study should be performed.
Detailed comments:
Latest works are missing from this survey, such as:
Wegner, F.K., Plagwitz, L., Doldi, F., Ellermann, C., Willy, K., Wolfes, J., Sandmann, S., Varghese, J. and Eckardt, L., 2022. Machine learning in the detection and management of atrial fibrillation. Clinical Research in Cardiology, 111(9), pp.1010-1017.
Król-Józaga, B., 2022. Atrial fibrillation detection using convolutional neural networks on 2-dimensional representation of ECG signal. Biomedical Signal Processing and Control, 74, p.103470.
Chen, Y., Zhang, C., Liu, C., Wang, Y. and Wan, X., 2022. Atrial Fibrillation Detection Using a Feedforward Neural Network. Journal of Medical and Biological Engineering, 42(1), pp.63-73.
Nankani, D. and Baruah, R.D., 2022, July. Atrial Fibrillation Classification and Prediction Explanation using Transformer Neural Network. In 2022 International Joint Conference on Neural Networks (IJCNN) (pp. 01-08). IEEE.
p.9 It is a very long and overly detailed explanation of basic and well-known concepts.
Section 4 - there is no need to describe classifiers that are not used in this work. And those that are used in this work should have implementation details specified.
Section 4.4 Again, there is no need in such details. Alas, F1 formula is wrong (it should have a beta parameter and then beta=2 is chosen, usually).
Table 3: The paper
Chen, Y., Zhang, C., Liu, C., Wang, Y. and Wan, X., 2022. Atrial Fibrillation Detection Using a Feedforward Neural Network. Journal of Medical and Biological Engineering, 42(1), pp.63-73.
reports detection sensitivity of >0.99 on this dataset.
Proposed methods are pretty basic and they should be compared to modern neural methods such as this one.
Tables 4-5 Same comment.
p.15 When results are so close numerically, please report statistical significance. Which one of the models are statistically the same?
Reviewer 2 Report
1 The structures should be descripted in the Introduction part.
2 The abstract, conclusion, and future work sections are too lengthy, and the authors should concise these parts.
3 The experimental conclusions should be more academic.
4 What is the meaning of "irregularly irregular"?
5 The methods in your experiments (Table 3) should be cited.
6 The English should be polished carefully.
7 The structures of your manuscript should be adjusted, and you can refer to other papers in Sensors.
8 The figures should be modified to enhance the readability.
Round 2
Reviewer 1 Report
My prior comments were not addressed in a satisfactory manner.
The main issue is - comparing results to one or more existing top methods, in the same setting.
Some claims in authors' response are confusing or wrong:
1) CNNs are the fastest NNs and are applied to ECG and EEG signals in multiple works, including online anomaly detection problems.
2) If you claim that your method's advantage is its runtime, then those runtimes should be reported TOGETHER with runtimes of competing methods in the same setting. Only then such a claim should be accepted.
3) There is a misunderstanding on authors' part of what is a statistical significance test. Authors should compare the output of their suggested methods (and baselines and toplines that should be added) using two-tailed pairwise Pearson's test and report significance according to the p-value.
4) Ablation study should be performed. It is a standard practice for all ML approached, neural or not.
Reviewer 2 Report
All my comments have been addressed, and I suggest that this revised version can be accepted.
